# Determinants of Public Health Personnel Spending in Spain

**DOI:** 10.3390/ijerph20054024

**Published:** 2023-02-23

**Authors:** Elena Puerto-Casasnovas, Jorge Galiana-Richart, Maria Paola Mastrantonio-Ramos, Francisco López-Muñoz, Alfredo Rocafort-Nicolau

**Affiliations:** 1Departamento de Empresa, Facultad de Economía y Empresa, Universitat de Barcelona, Diagonal 690-696, 08034 Barcelona, Spain; 2Departamento de Empresa, Facultad de Economía y Empresa, Universidad Autónoma de Barcelona, 08193 Bellaterra, Spain; 3Departamento de Contabilidad y Finanzas, EAE Business School, C/d’Aragó, 55, 08015 Barcelona, Spain; 4Facultad de Ciencias de la Salud, Universidad Camilo José Cela, Urb. Villafranca del Castillo, Calle Castillo de Alarcón 49, Villanueva de la Cañada, 28692 Madrid, Spain; 5Departamento de Contabilidad y Finanzas, La Salle, Universitat Ramon Llull, Carrer de Sant Joan de la Salle 42, 08022 Barcelona, Spain; 6Departamento de Economía, EUNCET Business School, 08225 Barcelona, Spain; 7Unidad de Neuropsicofarmacología, Instituto de Investigación Hospital 12 de Octubre (i+12), Avda. de Córdoba s/n, 28041 Madrid, Spain; 8Portucalense Institute of Neuropsychology and Cognitive and Behavioural Neurosciences (INPP), Universidade Portucalense, Rua Dr. António Bernardino de Almeida 541, 4200-072 Porto, Portugal; 9Red Temática de Investigación Cooperativa en Salud (RETICS), Red de Conductas Adictivas, Instituto de Salud Carlos III, MICINN y FEDER, 28029 Madrid, Spain

**Keywords:** public health expenses, public expenditures, GDP, public health personnel expenditures, Spanish health system, demographic factors

## Abstract

Public health is funded with government funds gathered from tax revenues, whether national, provincial or municipal. The health system therefore suffers during economic crisis periods, whether due to disinvestment, loss of purchasing power among health care personnel or the decrease in the number of professionals. This worsens the situation, as it is necessary to cover the needs of an increasingly elderly population and with a longer life expectancy at birth. The present study intends to show a model which explains the determination of the “Public Health Personnel Expenditure” in Spain for a determined period. A multiple linear regression model was applied to the period including the years 1980–2021. Macroeconomic and demographic variables were analyzed to explain the dependent variable. Variation in health personnel expenditure: “We included those variables which presented a high or very high correlation above r > 0.6. The variables which explain the behavior of Variation in health personnel expenditure”. It was a determining factor in the present study to consider that the variables with the greatest repercussions on health policy were mainly macroeconomic variables rather than demographic variables, with the only significant demographic variable that had a specific weight lower than macroeconomic variables being “Birth Rate”. In this sense, the contribution made to the scientific literature is to establish an explanatory model so that public policy managers and states in particular can consider it in their public spending policies, bearing in mind that health expenditures in a Beveridge-style health system, as Spain has, are paid with funds drawn from tax revenues.

## 1. Introduction

The World Health Organization defines the health system as “Such as all activities whose main priorities include promoting, restoring or maintaining health” (WHO) [1].

According to Fernández et al. [2], the public health sector is a fundamental pillar in the economic structure of any country, both because of the high direct and indirect investment it generates, and because of its social relevance [3], with its complexity and specificity as essential traits.

Referring to specificity, we must highlight the multidisciplinary and multicultural character of the public health sector [4]. Regarding its multidisciplinary nature, we can highlight that the health sector is constituted by different medical and clinical specialties, in both the public and private sectors. For multiculturality, each country has its own health policies and strategies, as well as micro-level management, i.e., regarding public or private hospital management, and reflecting the different social sectors involved in them [5]. According to Rodríguez Sánchez et al. [6], many of the studies conducted up to the present day show a correlation between health expenditures and the level of health attained by a national population [7,8,9].

If we consider the Spanish state, the health care system it currently has and which is the objective of the present study is articulated in response to the General Health Law of 14 April 1986, and the principal characteristics of “universality” and “gratuity” in its services, since all citizens in Spain have guaranteed health care [10].

The Spanish health care system lies within the framework of the Beveridge health care model [11,12], and is financed by public funds. In other words, it depends on the general budget of the state. These public funds come from income gathered via taxes, whether at the national, provincial or municipal level, thus accepting parallel and complementary coexistence with private health care.

According to data from the National Health System of Spain [13], public health care represents 71% of all health care expenditures in Spain, and although this percentage is high, as indicated by Martín and López del Amo [14], public resources must be increased to be able to cover the increasing needs of citizens with ever-longer lifespans [15].

In this vein, life expectancy in Spain is among the highest within the European Union at 83.4 years, compared to the EU median of 80.9 years. The social inequalities surrounding life expectancy are less pronounced than in other countries [16].

Per capita health care spending during the 2010s in Spain was about 15% below the EU average at 8.9% of GDP, compared to the EU median of 9.8% [17]. Apart from the greying of the population [18] we should mention that, according to Oliva et al. [19], the Spanish health system has been hit hard by an economic crisis with a drop in the number of professionals, public expenditures and investment. This crisis led, during the period of 2009–2013 [20], to a drop in health care expenditures of 12%, while between 2009 and 2016 the overall Spanish health care budget decreased by 4% [21,22], thus showing the need to find new measures to guarantee future sustainability for the system [23]. This is why, according to Villalobos Hidalgo [24], reform and innovation are needed in health system organization and management. Blanco et al. [25] indicated that there are already studies which try to estimate future public expenditures, specifically health care costs [26,27,28,29,30,31].

The current global crisis has reinforced the importance of having a powerful public health system in order to face new epidemics in general, and new pandemics arising from globalization in particular, although Spain in the last ten years has seen public health care resources dwindle [19].

According to Adhanom [32], COVID-19 showed the fragility of many health systems and services worldwide, and it is obligating countries to make difficult decisions about the best way to satisfy their citizens’ needs.

Beginning to read about current events in Spanish health care is depressing, according to Miranda [33], with constant reforms, decrees and laws leading to cuts and rapid, continuous change. For public health personnel, it is easy to feel feeble and irrelevant when facing the managers of the Spanish health system, even though with the crisis caused by the pandemic, these acquire a new dimension, becoming a new line of territorial defense.

According to the Spanish Health System Report, by Bernal-Delgado et al. [34], there was a 12.2% reduction in public health expenditures between 2009 and 2015, for a value of EUR 3.671 billion and −0.6% of gross domestic product (GDP) as a consequence, among other factors, of decreased personnel and investment expenditures. Personnel costs were cut by EUR 2.433 billion (8.0%), which shows the decrease in salaries and worker numbers.

In this sense, Muthuri et al. [35], referred to health professionals’ motivation as an important factor in health policy development, with job satisfaction as a key consideration.

In this vein, Blázquez-Fernández et al. [36] analyzed the main areas of public health care costs, concluding that there were territorial imbalances regarding the investment between the different autonomous communities (AC), so that increased health care expenditures were reflected in better health results. Similarly, Peña-Longobardo et al. [37] maintained that reduced health care results cause copious labor losses, caused by lost productivity, whether due to worker absenteeism, temporary or permanent work incapacity, or premature death. In this sense, Martínez-Pérez et al. [38], talking in relation to the wellbeing of the general population, maintained that quality of life is linked with health.

Spanish public health care, as Miranda [33] went on to say, has various elements which can be corrected and improved, while health care personnel as a collective are familiar with the complexities of the system, and should be able to come together and build consensus with managers and politicians to join together all the interests, knowledge and experience needed to apply, administer and maintain the system.

Thus, the present study is centered on public health personnel, defining which variables determine the behavior of “Public Health Expenditures” linked to public entities, in order to develop an explanatory model allowing for calculations of these costs for a determined period, and allowing its managers and actors to foresee variations in their purchasing power.

### Descriptive Analysis

The most representative section of health costs is personnel expenditures [39], and in countries with public health care paid for with public tax funds, health personnel payments (and thus health personnel) will be related to GDP.

In this sense, according to the World Health Organization [40], COVID-19 originated in Wuhan (China) in December 2019, and, from this point, has extended to almost the entire world [41]. It has infected millions of people and generated a pandemic with very few comparisons in history [42,43], with an economic effect of gigantic dimensions in all countries worldwide, since all economies are interrelated due to globalization [44]. Spain has been among the most affected countries [44,45], in spite of ranking 15th in the global health security index [46].

Governments have reacted to this crisis in more or less diligent ways, establishing new health policies [47] and decrees restricting population movements which have all generated impacts, certainly unfavorable ones, on their countries’ macroeconomic balances. According to the Bank of Spain [48], a domestic impact has arisen due to the following reasons: important rise in health care expenditures; a rise in extraordinary government spending, due to the implementation of economic aid measures for both individuals and companies and other entities. In Spain, we can refer to the application of Royal Decree 463/2020 [49], of 14 March, which declared a state for managing the health crisis caused by COVID-19, which includes, among other measures, processing files for temporary employment regulation (FTER), which increases the unemployment payments made by the Social Security administration along with assistance measures for the autonomous. This includes assimilation of personnel with dependency contracts, allowing them to receive unemployment payments; delays/moratoria on companies’ Social Security and state tax payments; guarantors allowing quicker and cheaper business financing; and aid intended to facilitate, for both businesses and individuals, the postponement of rents, mortgage payments and supply payments.

All the aforementioned factors led to reduced GDP as most companies’ business activities ceased completely, because of the total lockdowns put in place by Spanish government decree and the GDP drop due to partial paralysis of different economic activities [50].

In this sense, at the beginning of the crisis during March 2020, the consultancy firm Statista [51] went ahead with a study through which it put forth four possible scenarios reflecting the impact that isolating the population and freezing business activity could have on Spanish GDP. The four scenarios presented were as follows. The first scenario was 30 days of restrictions, 100% drop in retail, hospitality, transport and leisure activity and 20% for other areas. The second scenario was 30 days of restrictions, 30% drop in retail, hospitality, transport and leisure activity and 10% for other areas; the third scenario had 15 days of restrictions, 100% drop in retail, hospitality, transport and leisure activity and 20% for other areas; and the last scenario had 15 days of restrictions, 30% drop in retail, hospitality, transport and leisure activity and 10% for other areas.

In this sense, we can already affirm that the COVID-19 crisis has led the Spanish economy to register an 11% GDP drop in 2020, although the fourth trimester of the year saw positive numbers, with quarterly growth of 0.4% [52], and many more restrictions than previously foreseen.

## 2. Aims of the Study

The aim of the study was to develop an explanatory model to calculate “Public Health Personnel Expenditure” in Spain for a given period.

In this study, to approach this explanatory model we considered the timeframe of 1980–2021.

The number of observations refers to the study timeframe of 41 years. For the consideration of different variables, we have employed the studies conducted by Marchildon et al. (2011) [53], Santric-Milicevic et at. [39] and Hernández-Peña et al. [54] which have been considered for all the variables under scrutiny, which are the following: public health personnel spending, overall public health expenditures, gross domestic product, and total public spending, births, birth rate, fertility rate, deaths, mortality rate, average age of Spanish population, life expectancy at birth, public health expenditure as a percentage of total health expenditure, public expenditure as a percentage of GDP and public health expenditure per capita.

We studied the determining factors for health personnel spending in Spain, considering that this system is highly decentralized. Data originated from the National Statistics Institute [55].

For the multiple linear regression model, we chose the macroeconomic variables and demographic variables.

This study aimed to analyze the behavior of health personnel spending, in order to be useful to health managers and to state authorities when making decisions.

## 3. Methods

### Initial Study for Calculating COVID-19 Impact on Spanish GDP

A variable analysis was conducted using empirical evidence from the data extracted from INE, as subsequently mentioned.

To establish a mathematical model which explained the behavior of “Health Personnel Expenses” linked to public entities, we considered the period from 1980 to 2021 with the objective of identifying explanatory macroeconomic and demographic variables.

We analyzed the time series (1980–2021), during which a consistent pattern was detected where “Public Health Personnel Expenses” adjusted to GDP variations from year to year. The GDP growth generated during 2019 should thus be applied to “Public Health Personnel Expenditures” in 2020, thereby generating increased purchasing power for said personnel and/or personnel increase.

However, COVID-19 caused the cessation of business activity imposed by the Spanish government, an unforeseen increase in health care expenditures and a rise in extraordinary expenditures due to the implementation of economic aid measures for businesses, individuals and other entities. This meant that the “Public Health Care Personnel” area was negatively impacted due to the direct relation it has with GDP variations, as the latter number saw an initial variation calculation ranging from −2.83% to 0.83%.

Using data from the National Statistics Institute (INE) [56] on the evolution of Spanish GDP and public spending from 1980 to 2021, we can see that, while the GDP tended to increase after 2014 (Figure 1), the rise in public spending did not follow the same trend (Figure 2). This was due to cost-containing measures implemented during this period by the Spanish national government and autonomous communities (AC), just as was reflected in the Annual National Health System Report [57], which indicated that “since 2009, when the historic upward trend of public health care expenditures stopped, public administrations’ expenditures have decreased by 6.0%, giving rise to extraordinary public deficit reduction measures adopted since May 2010 due to the economic crisis”.

The evolution of “Public Health Care Expenditures” and “Expenditure on Health Personnel”, using monetary euros, does not subtract from the inflation effect. In spite of the variations which “Public Expenditures” saw during the 1980–2021 period, “Public Health Care Expenditures” remained virtually stable (Figure 3 and Figure 4).

Once the evolution of “Public Expenditures” has been analyzed, along with its relation to “Public Health Care Costs” and deflated GDP, based on INE data [58], we can study a more concrete parameter, such as “Public Health Personnel Expenditures”, which is related with “Public Health Care Costs” between 1980 and 2021, i.e., it is necessary to perform an econometric analysis to explain the variable “Health Personnel Expenditure Variation”, considering demographic variables.

Variations in both magnitudes followed similar trends, i.e., when there was an increase in “Health Care Expenditures”, there was a rise in “Public Health Care Personnel Expenditures” (Figure 3 and Figure 4).

Taking the figures from the International Monetary Fund (IMF) [59], which established a Spanish GDP growth rate of 1.6% prior to the COVID-19 crisis, an estimate was obtained for a “pre-COVID-19 GDP” for the year 2020. By using this “pre-COVID-19 GDP” and applying the restrictions contemplated by Statista [51] in its four scenarios, we obtained a forecast of a “COVID-19 GDP” for 2020 for each scenario, which in the best of cases involved a monetary impact of COVID-19 on the GDP of EUR 9.552 billion, and in the worst case, of EUR 55.188 billion.

This showed that, in any of the four cases, the perspective of retributive variation for health care workers was negative, since they could face a purchasing power loss ranging between 0.90% and 1.78%. The estimation was made by using time data extracted from the INE, concerning GDP evolution, public spending and public health expenditures. An important part of the latter point was health personnel spending.

Considering that, with this research focus, the results obtained were not precise, but ranged within a wide bracket, the authors considered reorienting their focus towards a multiple linear regression model.

For the second study, the process and data analysis were completed via the IBM SPSS version 27.0 statistical packet for Windows, with macroeconomic variables including annual variation in public expenditures; annual health spending variation; annual GDP variation; and demographic variables including birth rate, death rate, median population age, population over 65 and annual population variation.

We initially performed a descriptive analysis of the study target variables (Table 1), including the normality analysis (Table 2).

In this case, we used the Shapiro–Wilk test, since the number of data points obtained was below 50.

Since the significance level was under 0.05, implying a lack of normality, the following variables were excluded from analyses: “Death Rate”, “Life Expectancy at Birth”, “Annual Population Variation” and “Percentage of Public Spending over GDP”.

The dependent variable was “Annual Variation in Health Personnel Expenditure” and the independent variables were “Birth Rate”, “Median Population Age”, “Population over 65 years old”, “Annual Public Spending Variation”, “Annual Health Expenditures Variation“ and “Annual GDP Variation”. We intended to know whether these dependent variables predicted the behavior of the variable “Health Personnel Expenditure Variation”.

A hierarchical regression model was used to identify more information about the dependent variables being used (Table 3).

The Durbin–Watson test score indicated an independence of errors (2.478) since the value was between 1 and 3.

For the multiple regression model tested, we can explain 80.4% of the variance in the dependent variable. However, models 4 through 6 explained the same percentage. This means that the variables included in models 5 and 6 did not contribute to explaining the model, i.e., the variables “Median Population Age” and “Population Over 65” can be withdrawn and not used.

## 4. Results

This study seeks to answer the following question: What are the determinants of public health personnel spending in Spain?

In the simple regression analysis conducted with the six models, we saw that the determination coefficient was 0.804, which shows that the variance proportion explaining “Health Personnel Expenditures” was very high.

Below is the ANOVA and the table of coefficients (Table 4).

The ANOVA for the regression model with four variables indicated that this significantly improved the prediction of the dependent variable (F:38.863, *p*-value < 0.001) (Table 5).

For the regression model coefficients, the t scores indicated that the variables considered significantly contributed to the predictive model. Simultaneously, the Inflated Variance Factor (FIV) indicated compliance with the non-multicollinearity assumption (values between 1.209 and 2.861).

The regression equation was as follows:Variation in health personnel spending = −0.41 + 0.126 × Public Spending Variation + 0.005 × Birth Rate + 0.645 × Annual health spending variation + 0.097 × Annual GDP variation.

With the equation resulting from the model, we can see that the variation in health personnel spending had explicative variables, and only one demographic variable, namely Birth Rate.

## 5. Discussion

To be able to answer the research question in the present study—what are the determinants for public health personnel expenditures in Spain?—we have borne in mind that personnel health outlays are one of the most important parts of all public health spending, with health personnel accounting for around 57% of all health costs [58,59].

Even after realizing that GDP determines the wealth of a country and that total public spending is related with GDP, it is currently hard to coherently estimate income due to the lack of available data, leading to deficient decision-making by health managers [60]. In this sense, the World Health Organization is making important efforts to gather and unify homogeneous data from all member states using data from the International Labor Organization [61]. WHO-CHOICE has also been used to perform salary cost estimations.

In the present study, the first analysis performed was considered to lack relevance, since its results were not highly concrete and lay within a bracket which was too wide to carry out reliable explanatory calculations on the variability of “Public Health Personnel Expenditures”, and therefore required econometrics to achieve the objective set forth in the present study.

In the second analysis, we used a regression model where the dependent variable was “Public Health Personnel Expenditure”, and the explanatory variables used were both macroeconomic and demographic.

We can thus affirm that the demographic variables have been significant for determining variation in “Public Health Personnel Expenditure“.

There are prior studies, such as the case of Muthuri et al. [35], where a systematic review was conducted to synthesize the determining factors for health workers, with the study centering on the East African community. There are also studies which attempt to predict the evolution of health care spending, bearing in mind the demographic impact, health status, death-related costs and some macroeconomic variables [62]. Along the same lines, Hitiris et al. [60,63] concluded that the GDP was an important health spending determinant. Braendle et al. [64] dealt with public spending determinants within Switzerland during the 1970–2012 timeframe. Serje et al. [65] studied the estimated salaries of health workers worldwide, determining that incomes showed a negative correlation with GDP, and reaching the conclusion that countries with lower incomes paid health workers relatively better than higher-income countries.

In line with the previous points, although there are prior studies relating health care spending with GDP, there are no available studies relating it with public health care spending under the Beveridge health model [60].

This study is not without limitations, since available data are limited, and it is not homogenized in various occasions. We also did not consider tax collection variables, budget deficits, public employment or the role of unions due to data homogenization problems, and the analysis covers a time span of 41 years, which is the period 1980–2021, since earlier dates for some variables had no data available.

For future research lines, we suggest considering more countries inside and outside the European Union, since there are various health care models available, such as the Bismarck, Beveridge, liberal and socialist types. As a function of the reigning model, the results could be different. To this end, there is an important effort being made to homogenize data.

One relevant data point to bear in mind would be considering the sustainable development objectives (SDO), which in a broader timeframe present an estimated deficit of 18 million health workers by the year 2030 [66].

## 6. Conclusions

### 6.1. Implications for Health Care Provision and Use

Due to progress achieved in increasing life expectancy and reducing mortality rates, it is ultimately essential for health policy makers to increase investment in health, since otherwise all the achievements up to now could revert, giving rise to the high costs associated with illness.

Increased life expectancy is associated with more diseases, and thus with increased health spending, which must be considered by health policy managers. Spain is one of the countries with the highest life expectancy at birth; in 2021, this stood at 83.06 years.

With the results obtained, it should be mentioned that health personnel spending variation depends largely on macroeconomic variables rather than demographic variables, since the only demographic variable we have considered is the birth rate. Out of all the variables considered in the model, it is also the one with the least effect on the dependent variable.

### 6.2. Implications for Health Policies

We know that personnel spending is one of the main outlays for the public health system. Therefore, given the results presented herein, to carry out a good policy for the resources aimed at public health personnel expenditures, we can expect that national economic activity is favorable in terms of GDP and wealth creation, since we have shown that most determining factors for health personnel expenditure variation are macroeconomic variables. This is especially significant, since the present study indicates that demographic variables are not considered for establishing health policies.

Economic activity in different countries and the decisions taken regarding public spending policies both ultimately have significant effects on health personnel expenditures, which, as the life expectancy rises and the population ages, mean that more and more of the population have multiple pathologies and will have greater health service needs.

States will have to be conscious of the importance of being able to cover their populations’ needs, especially since their characteristics will vary over time, and GDP and public expenditures will be a determining factor for all states.

### 6.3. Implications for Further Research

The future lines of this work will include considering the impact of COVID-19, especially considering the various pathologies which can arise due to the pandemic and the medium- and long-term effects it will have on the population. This can lead to re-founding of health policies in the future, both in matters referring to public spending and the cost represented by health care personnel. It will also be interesting to carry out this study between different autonomous communities (AC), bearing in mind that significant differences exist in their public health systems.

Due to the importance of various different studies where health personnel are responsible for about 57% of all health costs, future studies could consider how inflation will affect health costs, and specifically personnel costs. Health policy managers currently lack consistent data about income estimations, leading to major data shortfalls within global policies. In this sense, the efforts of the World Health Organization to gather data from all member states using International Labor Organization data WHO-CHOICE have also been used to make salary cost estimates.

In the face of this situation, the present study contributes to health policy managers being able to consider possible future actions to take if they intend to maintain quality public health, and not trigger a significant migration out of the health care workforce.

## Figures and Tables

**Figure 1 ijerph-20-04024-f001:**
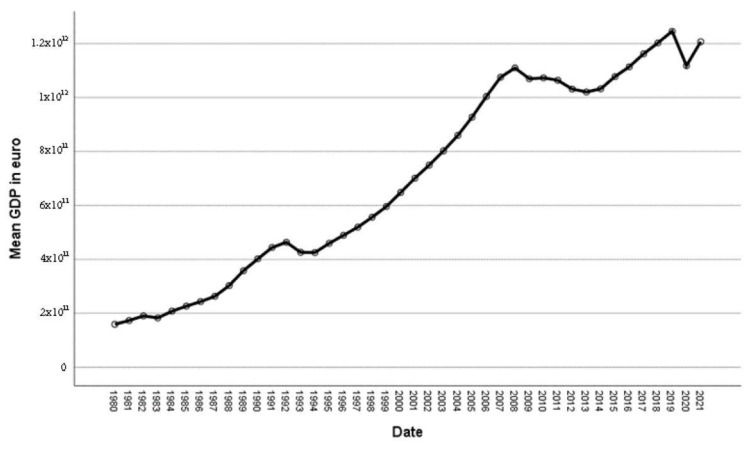
Evolution of GDP. Source: National Statistics Institute (INE) [55].

**Figure 2 ijerph-20-04024-f002:**
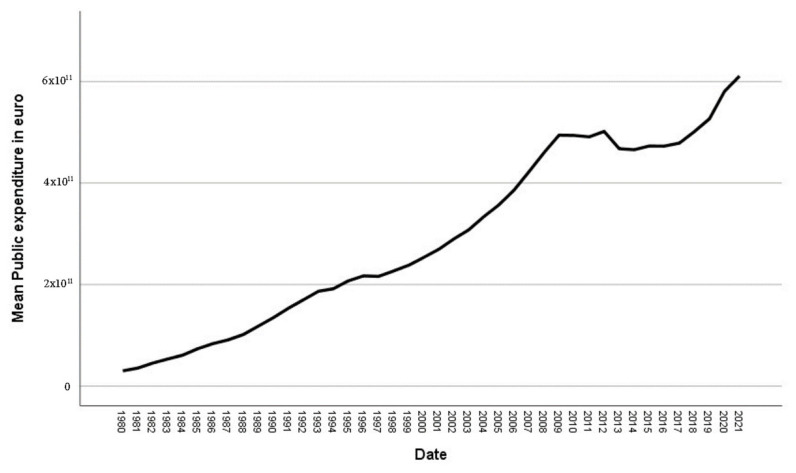
Evolution of public expenditure. Source: National Statistics Institute (INE) [55].

**Figure 3 ijerph-20-04024-f003:**
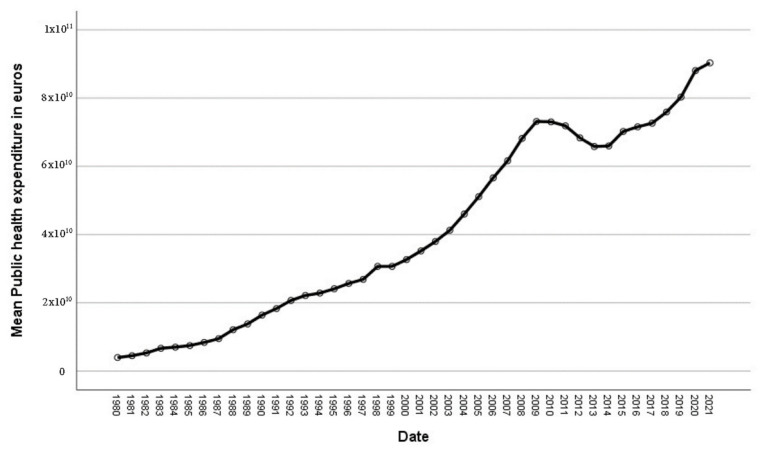
Evolution of public health expenditure. Source: National Statistics Institute (INE) [55].

**Figure 4 ijerph-20-04024-f004:**
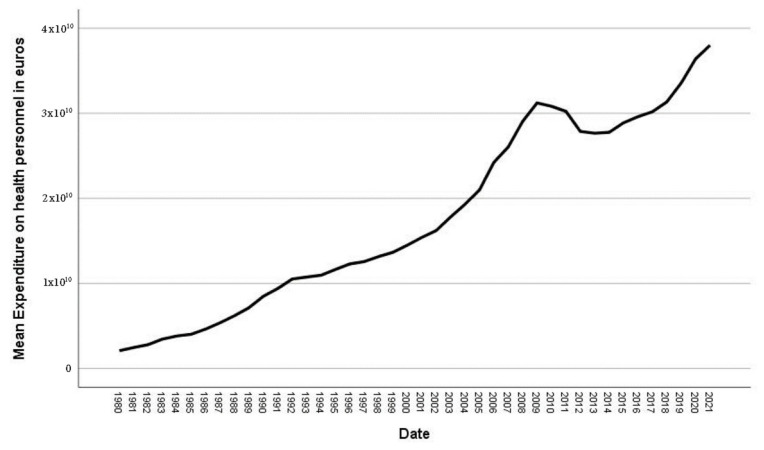
Evolution expenditure on health personnel. Source: National Statistics Institute (INE) [55].

**Table 1 ijerph-20-04024-t001:** Descriptive statistics.

	N	Minimum	Maximum	Mean	STD Deviation
Birth rate	42	7.1000	15.2200	10.208095	1.6857715
Death rate	42	7.5200	10.4000	8.546905	0.5247693
Median age of population	42	26.1900	49.7800	39.110476	3.9757717
Life expectancy at birth	42	75.44	83.58	79.5607	2.58854
Population over 65	42	11.24	20.08	15.8090	2.57171
Annual population variance	42	−0.003713	0.019343	0.00564207	0.005842911
Annual health personnel expenditure variation	42	−0.078130	0.234810	0.07753357	0.065311194
Annual public spending variation	41	−0.067849	0.278202	0.07825466	0.066680494
Annual health spending variation	41	−0.049374	0.275106	0.08127524	0.069073167
Annual GDP variation	41	−0.102387	0.182401	0.05227666	0.058309309
% Public spending over GDP	42	0.039600	0.080000	0.05431667	0.010557968
Valid N (by list)	41				

**Table 2 ijerph-20-04024-t002:** Normality test.

	Shapiro–Wilk		
	Statistical	gl	Sig.
Birth rate	0.967	41	0.284
Death rate	0.945	41	0.048
Median population age	0.951	41	0.078
Life expectancy at birth	0.921	41	0.007
Population over 65 years old	0.949	41	0.066
Annual population variation	0.846	41	0.000
Annual health personnel spending variation	0.983	41	0.776
Annual public expenditure variation	0.969	41	0.328
Annual health expenditure variation	0.963	41	0.197
Annual GDP variation	0.954	41	0.096
% Public spending over GDP	0.945	41	0.045

**Table 3 ijerph-20-04024-t003:** Model summary.

Model	R	R Squared	Adjusted R^2^	Standard Estimation Error	Durbin–Watson
1	0.731 ^a^	0.534	0.522	0.044529030	
2	0.738 ^b^	0.545	0.521	0.044564984	
3	0.893 ^c^	0.797	0.781	0.030139396	
4	0.897 ^d^	0.804	0.782	0.030068912	
5	0.897 ^e^	0.804	0.776	0.030489495	
6	0.897 ^f^	0.804	0.770	0.030909299	2.478

^a^ Predictive variables: (Constant), Annual Public Spending Variation. ^b^ Predictive variables: (Constant), Annual Public Spending Variation, Birth Rate. ^c^ Predictive variables: (Constant), Annual Public Spending Variation, Birth Rate, Annual Health Expenditure Variation. ^d^ Predictive variables: (Constant), Annual Public Spending Variation, Birth Rate, Annual Health Expenditure Variation, Annual GDP Variation. ^e^ Predictive variables: (Constant), Annual Public Spending Variation, Birth Rate, Annual Health Expenditure Variation, Annual GDP Variation, Median Population Age. ^f^ Predictive variables: (Constant), Annual Public Spending Variation, Birth Rate, Annual Health Expenditure Variation, Annual GDP Variation, Median Population Age, Population Over 65.

**Table 4 ijerph-20-04024-t004:** ANOVA ^e^.

Model		Sum of Squares	df	Mean Square	F	Sig.
1	Regression	0.089	1	0.089	44.652	0.000 ^a^
	Residual	0.077	39	0.002		
	Total	0.166	40			
2	Regression	0.090	2	0.045	22.758	0.000 ^b^
	Residual	0.075	38	0.002		
	Total	0.166	40			
3	Regression	0.132	3	0.044	48.532	0.000 ^c^
	Residual	0.034	37	0.001		
	Total	0.166	40			
4	Regression	0.133	4	0.033	36.863	0.000 ^d^
	Residual	0.033	36	0.001		
	Total	0.166	40			

^a^ Dependent variable: Expenditures PERSONNEL SPAIN; ^b^ Predictors: (Constant), GDP in €; ^c^ Predictors: (Constant), GDP in €, Public Expenditures. ^d^ Predictive variables: (Constant), Annual Public Spending Variation, Birth Rate, Annual Health Expenditure Variation, Annual GDP variation. ^e^ Dependent variable: Annual Health Personnel Expenditure Variation.

**Table 5 ijerph-20-04024-t005:** Coefficients ^a^.

Model		Unstandardized Coefficients		Standardized Coefficients	t	Sig.	Statistics of Collinearity	
		B	Std. Error.	Beta			Tolerance	FIV
4	(Constant)	−0.041	0.039		−1.053	0.299		
	Annual Public Spending Variation	0.126	0.121	0.130	1.043	0.037	0.350	2.861
	Birth Rate	0.005	0.004	0.114	1.120	0.027	0.526	1.902
	Annual Health Spending Variation	0.645	0.098	0.692	6.563	0.000	0.490	2.040
	Annual GDP Variation	0.097	0.090	0.088	1.083	0.028	0.827	1.209

^a^ Dependent variable: Annual Public Health Personnel Expenditure Variation.

## Data Availability

Not applicable.

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
