# Peer review of "Determinants of Public Health Personnel Spending in Spain"

_ijerph, 2023, doi:10.3390/ijerph20054024_

Round 1

Reviewer 1 Report (Previous Reviewer 1)

Dear editor,

The article is not yet ready for publication, despite my appreciation for the efforts made.

The paper needs a better structure: a paragraph devoted to the descriptive analysis, one to the relevant empirical literature, one to the data and methods, and finally one to the results. Currently the results paragraph combines a descriptive analysis with the results. Seemingly the paragraph on method has a descriptive side. The descriptive analysis should be put all together in one paragraph. The paragraph on method should only present the empirical approach. Still it is not clear the method used for computing the retributive variation of health personnel because of COVID ( Row 240-242). Everything should clearly spelled out about the assumptions made and how.

Without providing the theoretical justifications for their selections, the authors chose specific variables (rows 176–179). The authors should use the relevant empirical literature to guide the selection of variables to be considered as regressors if the article's goal is to explain the factors that influence the cost of health personnel. Why, for instance, does not population growth affect the number of employees and subsequently the cost of health personnel? Why did they take into account the median age of the Spanish population but not the proportion of people over 85 to  evaluate the effects of ageing?

 In this respect they could refer, among others, to studies like :

Vaughan, K., Kok, M.C., Witter, S. et al. Costs and cost-effectiveness of community health workers: evidence from a literature review. Hum Resour Health 13, 71 (2015).

Gregory Marchildon and Livio Di Matteo (2011) Health Care Cost Drivers: The Facts. CIHI Report

Matthias Wismar, Claudia B Maier, Anna Sagan and Irene A Glinos, (2018) Developments in Europe's health workforce: addressing the conundrums ; Journal. Eurohealth, 24 (‎2)‎: 38 – 42

Santric-Milicevic, M., Vasic, V. & Terzic-Supic, Z. Do health care workforce, population, and service provision significantly contribute to the total health expenditure? An econometric analysis of Serbia. Hum Resour Health 14, 50 (2016). https://doi.org/10.1186/s12960-016-0146-3

Serje, J., Bertram, M.Y., Brindley, C. et al. Global health worker salary estimates: an econometric analysis of global earnings data. Cost Eff Resour Alloc 16, 10 (2018). https://doi.org/10.1186/s12962-018-0093-z

Hernandez-Peña P, Poullier JP, Van Mosseveld CJ, Van de Maele N, Cherilova V, Indikadahena C, Lie G, Tan-Torres T, Evans DB. Health worker remuneration in WHO Member States. Bull World Health Organ. 2013;91:808–15.

Hernandez-Peña P, Poullier JP, Van Mosseveld CJ, Van de Maele N, Cherilova V, Indikadahena C, Lie G, Tan-Torres T, Evans DB. Health worker remuneration in WHO Member States. Bull World Health Organ. 2013 Nov 1;91(11):808-15

Is the paragraph "Descriptive analysis of Covid-19 and Spanish GDP" really necessary given the purpose of the paper?

At row 273 certain variables are not considered as non normal in the distribution. Linear regression analysis does not assume normality for the predictors. The normality of the residuals is the one to be checked which is not presented. Still no test for omitted variables or heteroschedasticity is also presented.

Since regressors with high correlation are only presented (rows 290–292), it is generally unclear which other variables are taken into account. What other variables has the correlation been tested for? For those excluded because of the normality issue?

The author uses same variables expressed in different terms so the multicollinearity test would be needed. The author refers to the results of the multicollinearity in the replies to the referees but I don’t see the table in the text. It should be put there in order to give a proof of what said.

Then the Analaysis of variance is presented without explaining why it is needed.

It would be preferable to present the results of the regression by introducing a variable at a time  so that the reader can see the contribution made by each regressor?

In general the article should be carefully read as there are words not related to the context like correlations before the table of the descriptive statistics,  7 models at row 331 while there is only one single model.

Something is missing at row 222  but in general the entire paragraph is not really clear.

Specifically it is not clear what the author does in order to take care of inflation. 

Author Response

This manuscript is a resubmission of an earlier submission. The following is a list of the peer review reports and author responses from that submission.

Round 1

Reviewer 1 Report

Dear Editor,

The issues raised in the first round of comments have not been addressed in the revised version of the paper.

The economic analysis remains insufficiently robust. The sample size under consideration is insufficient. A panel analysis involving other European countries or a longer series should be provided to address this issue. 

There is, however, no correlation matrix of public health personnel expenditure with all of the determinants considered. It is unclear what other variables were considered in addition to those used in the model. The literature review was not used to identify additional regressors. Despite the fact that they were required, no post estimation tests for omitted variables or model specification were provided.

I still don't see any analysis of the model's predictive accuracy.

The para starting at row 220 up to 226 is not clear. The authors should explain what they did, how they did it, and clarify better the assumptions made.   

Minor revisions: the heading of Fig. 5 is not correct.

Reviewer 2 Report

The issue of Determinants of Public Health Personnel Spending in Spain was addressed in the manuscript. Due to the experience of the pandemic, the subject of health protection is quite often present in the scientific literature.

The analysis covers many important issues for the public health care system. I evaluate this multifacetedness positively. However, there is a lack of proper reference to an issue that will have a significant impact on the future of healthcare. This process is the aging of the society and the challenges associated with it. This process will affect many aspects of the functioning of societies and economies of many countries. It also creates a number of challenges for healthcare systems. I think this thread should be added.

The entries in the figures should be corrected. The data presented in the figures come from official statistics. Therefore, there can be no entry: "source: authors".

The article lacks a solid, structured approach. As an example, you can include references to several bibliographic items in the article's limitations section. Limitations of the analysis should be a part where certain reservations and limits within which the research was conducted are shown. Assuming other frameworks or assumptions, the results could be different. Rather, it should not be a place where many references are cited.

The Conclusions section should also be a place where the authors summarize their analyzes and indicate their own recommendations. So there shouldn't be room to refer to multiple references. this is the case with the peer-reviewed manuscript. References should be rather limited in this section.

 Certain provisions should be corrected, as the standard slightly differs from the regime of a scientific article. I mean, for example, sentences starting with: "Figures 1 and 2 show ...."; "As Figures 3 and 4 show...".

In the references, a large part are items published in Spanish, which makes it difficult to assess the manuscript and the legitimacy of their use.

Reviewer 3 Report

The author has followed some of the suggestions but not in toto. The red marked comments are to be incorporated to strengthen the position of the authors. Hence, minor revisions are required on the red marked comments in the Review Report as attached.

Reviewer 4 Report

The paper is not suitable for pulication for the following reasons.

First, the paper has very weak motivations for study. As the paper describes, health care expenditure of Spain is financed by public funds and is thus dependent on the budget of the state. Government budget status is dependent on the national income on the whole. The regession model in page 10 is a simple econometric exercise that shows a rough picture of the effects of GDP on spending where it includes public health personnel expenditure (which is a part of health expenditure). The authors present motivations for the study in the paper, which, however, are neither  convincing motivations not making contributions in the literature. Besides, it is not very persuative, either, to focus only on the health personnel expenditure out of the health expenditure. Personnel expenditure can change with other (policy) factors such as (change in) salary schemes for the personnel (with roles played by trade unions), supply of staff, on-going digitalization process in the health care industry, and so on. Only focusing on the GDP with the health personnel expenditure is not catching any intesest. 

Second, as mentioned already above, public budget is closely associated with GDP. Thse two very closely related variables, GDP and public expenditure are included in one regression model, which is not convincing.  

The papaer allocates a good share in discussion on the effects of Corona on health personnel expenditure, which is not clearly understandable. The pandemic has made many countries to increase spending on public health by purchasing massive vaccianes and reallocating/mobilizing personnel for enabling vaccinations/boosters, among others, which causes increasing deficit in public finance. A fall in GDP in year 2020 and 2021 can't be a pleasible explanatory factor for a change health personnel expenditure. Here, it reaffrims that the regression model where health personnel expenditure in explained by GDP and the health care expenditure is not very appealing both from economics and econometrics perspectives.
